# Effectiveness of Tri-Kaysorn-Mas Extract in Ameliorating Cognitive-like Behavior Deficits in Ovariectomized Mice via Activation of Multiple Mechanisms

**DOI:** 10.3390/ph17091182

**Published:** 2024-09-08

**Authors:** Abdulwaris Mading, Yutthana Chotritthirong, Yaowared Chulikhit, Supawadee Daodee, Chantana Boonyarat, Charinya Khamphukdee, Wanida Sukketsiri, Pakakrong Kwankhao, Supaporn Pitiporn, Orawan Monthakantirat

**Affiliations:** 1Graduate School of Pharmaceutical Sciences, Khon Kaen University, Khon Kaen 40002, Thailand; abdulwaris.m@kkumail.com (A.M.); yutthana_ch@kkumail.com (Y.C.); 2Division of Pharmaceutical Chemistry, Faculty of Pharmaceutical Sciences, Khon Kaen University, Khon Kaen 40002, Thailand; yaosum@kku.ac.th (Y.C.); csupawad@kku.ac.th (S.D.); chaboo@kku.ac.th (C.B.); 3Division of Pharmacognosy and Toxicology, Faculty of Pharmaceutical Sciences, Kaen University, Khon Kaen 40002, Thailand; charkh@kku.ac.th; 4Division of Health and Applied Sciences, Faculty of Science, Prince of Songkla University, Songkhla 90110, Thailand; wanida.su@psu.ac.th; 5Department of Pharmacy, Chao Phya Abhaibhubejhr Hospital, Ministry of Public Health, Prachinburi 25000, Thailand; pakakrong2@gmail.com (P.K.); spitiporn@yahoo.com (S.P.)

**Keywords:** Tri-Kaysorn-Mas, learning and memory, neurodegeneration, antioxidant, ovariectomy

## Abstract

Postmenopausal women have a higher probability of experiencing cognitive alterations compared to men, suggesting that the decline in female hormones may contribute to cognitive deterioration. Thailand traditionally uses Tri-Kaysorn-Mas (TKM), a blend of three medicinal herbs, as a tonic to stimulate appetite and relieve dyspepsia. Due to its antioxidant and anti-acetylcholinesterase activities, we investigated the effects of TKM (50 and 100 mg/kg/day, *p.o*., for 8 weeks) on cognitive deficits and their underlying causes in an ovariectomized (OVX) mouse model of menopause. OVX mice showed cognitive impairment in the Y-maze, novel object recognition task (NORT), and Morris water maze (MWM) behavioral tests, along with atrophic changes to the uterus, altered levels of serum 17β-estradiol, and down-regulated expression of estrogen receptors (ERα and ERβ). These behavioral effects were reversed by TKM. TKM decreased malondialdehyde (MDA) levels and mitigated oxidative stress in the brain by enhancing the activity of superoxide dismutase (SOD) and catalase (CAT) and by up-regulating the antioxidant-related gene Nrf2 while down-regulating Keap1. TKM also counteracted OVX-induced neurodegeneration by enhancing the expression of the neurogenesis-related genes BDNF and CREB. The results indicate that TKM extract alleviates oxidative brain damage and neurodegeneration while enhancing cognitive behavior in OVX mice, significantly improving cognitive deficiencies related to menopause/ovariectomy through multiple targets.

## 1. Introduction

Dementia affects an estimated global population of over 50 million individuals, with Alzheimer’s disease being the predominant underlying cause. This condition ranks as the fifth most prevalent cause of mortality and accounts for 28.8 million disability-adjusted life years. Neuropathological alterations associated with Alzheimer’s disease can be detected in the brain several decades before the manifestation of Alzheimer’s dementia, possibly accompanied by modest cognitive deterioration [1]. Dementia is marked by cognitive impairment and is the most common neurodegenerative disease, with a greater incidence rate in women than men [2,3]. Estrogen plays a role in the regulation of the ovaries and uterus and in the sexual differentiation of several brain activities, including reproductive control and certain cognitive processes. The sharp decrease in circulating estrogen levels greatly impacts learning and memory function decline in aging humans, non-human primates, and rodent females [4]. These effects have been attributed to conventional genomic processes, which include the binding of hormones to receptors such as the estrogen receptors ERα and ERβ, alterations in gene transcription, and the initiation of organ-specific effects in both the brain and peripheral organs [5]. In Thai traditional medicine, the phrase “the blood will go, and the wind will come” is of great significance in understanding women’s health, particularly during menopause. “Blood will go” refers to the reduction in semha and pitta, indicating a decrease in the menstrual period and hormone levels. This leads to the aggravation of vata, or the arrival of wind, which acts as the dysfunction of the blood circulatory system. This aligns with the decline in estrogen levels and the resulting cognitive dysfunction during menopause. Moreover, increasing evidence suggests that gonadal hormones act as neurosteroids within the brain, thereby rapidly influencing cognition and various functions [6]. Hence, the reduction in estrogen levels due to post-menopause or ovariectomy may contribute to a lower occurrence of neurogenesis and neuroplasticity [7].

Research conducted in clinical settings has provided evidence indicating a notable prevalence of neurological disorders, such as Alzheimer’s disease (AD), among women who have reached menopause. The observed rise in incidence, a crucial area of study, has been linked to the decline in estrogen levels that occurs as a result of the menopause transition. For instance, previous studies have shown that estrogen can improve the non-amyloidogenic fragmentation of β-amyloid precursor protein (APP) due to the upregulation of α-secretase synthesis and facilitates the removal of β-amyloid (Aβ) by accelerating its uptake by astrocytes [8]. Estrogen has also been shown to diminish the accumulation of tau protein groups and enhance the production of acetylcholine transferase (ChAT) in the basal forebrain, thereby decelerating the cognitive deterioration associated with Alzheimer’s disease [9]. Nevertheless, it is important to acknowledge that other variables might contribute to the pathogenesis of Alzheimer’s disease, including mitochondrial dysfunction and oxidative stress [10,11]. The brain is classified as a postmitotic organ, meaning that its cells do not undergo cell division. This characteristic makes the brain more susceptible to oxidative damage by various factors, including its high reliance on oxygen, the presence of a substantial amount of easily oxidizable polyunsaturated fatty acids, an abundance of redox-active transition metal ions, and a relative deficiency in the antioxidant defense system [12]. Furthermore, it has been suggested that an alternation in the redox state of the CNS may play a role in influencing these pathological conditions [13]. Although hormone replacement therapy (HRT) is often used as the primary approach to mitigate or avoid the physiological changes associated with menopause, investigations have shown several detrimental consequences linked to prolonged HRT, including increased susceptibility to breast cancer, endometrial cancer, stroke, and thromboembolism. Considering these adverse effects, phytoestrogen might provide more advantages compared to standard HRT [14].

Tri-Kaysorn-Mas (TKM) is a traditional Thai medicine found in the Thai National Drug Information database. It consists of three medicinal herbs: *Aegel marmelos* fruit, *Jatropha multifida* bark, and *Nelumbo nucifera* stamen in a 1:1:1 ratio. Its infusion, decoction, and powder are used for appetite stimulation, dyspepsia treatment, and as a tonic. Previous investigations have shown that the Tri-Kaysorn-Mas formula has antioxidant and anti-acetylcholinesterase activity [15,16]. *A. marmelos* contains numerous phytochemicals at high levels, such as aegeline, scopoletin, and imperatorin, which exhibit phytoestrogen activity and neuroprotective effects [17,18,19]. *N. nuciferara* has diverse chemical groups, including flavonoids and alkaloids [20] like kaempferol, β-sitosterol, quercetin, and nuciferine, which show phytoestrogen activity and improve learning and spatial memory [21,22,23,24,25]. *J. multifida* has several chemical constituents, such as vitexin, and isovitexin, that have been reported to exhibit estrogenic activity and reduce cognitive impairment [26,27,28,29]. The therapeutic effectiveness of the Tri-Kaysorn-Mas formula may be attributed to the estrogenic and antioxidant properties exhibited by the individual herbs present in the recipe. However, there is no existing report on the effects of Tri-Kaysorn-Mas on cognitive and oxidative brain damage caused by ovariectomy. Hence, the primary aim of this research was to investigate the effects of TKM extract on improving cognitive impairment and its mechanisms of action on estrogen receptors, neurotrophic factors, and oxidative stress induced by ovariectomy in mice.

## 2. Results

### 2.1. Effects of TKM Extract on In Vitro Studies for Antioxidant Activities

The in vitro antioxidant activity of TKM extract was examined using oxygen radical absorbance capacity (ORAC) and superoxide anion (O_2_^•−^) radical scavenging activity. The results are shown in Table 1.

### 2.2. Effects of TKM Extract on OVX-Induced Cognitive-like Behavior

This study utilized the Y-maze test, NORT, and MWM test to evaluate the effectiveness of TKM extract at ameliorating learning and memory impairments caused by OVX. The Y-maze test showed that OVX mice receiving vehicle treatment exhibited a considerably lower percentage of alternation than the sham-operated group. OVX resulted in a decline in spatial or short-term memory. The percentage of alternation was significantly higher in the 100 mg/kg/day of TKM extract and 1 µg/kg/day of E2 groups compared to the OVX vehicle group (Figure 1) (for the detailed statistical analysis, see Appendix A).

The sham-operated control group mice showed a capacity to accurately distinguish between familiar and unfamiliar objects in the NORT, and the OVX mice that received the vehicle treatment exhibited an inability to recognize the unfamiliar object. OVX mice that were administered a daily dosage of 1 μg/kg of E2 (17β-estradiol) and 100 mg/kg of TKM extract enhanced their discrimination index performance. TKM extract exhibited a dose-dependent response in the NORT (Figure 2) (for the detailed statistical analysis, see Appendix A), with improved discrimination indices between 50 and 100 mg/kg/day dosages.

Mice in all treatment groups demonstrated a consistent decrease in the time spent locating the submerged platform over the five days of the training phase of the MWM, indicating a steady learning ability (Figure 3a) (for the detailed statistical analysis, see Appendix A). The sham-operated control mice showed a notable reduction in escape latency times compared to OVX mice treated with the vehicle on days 2, 3, 4, and 5. Furthermore, treatment with TKM extract and E2 for three weeks substantially decreased escape latency on days 2, 3, 4, and 5. Regarding the probe phase on day 6 (Figure 3b) (for the detailed statistical analysis, see Appendix A), this result suggests that OVX-induced learning and memory impairments. Administering E2 at a dosage of 1 μg/kg/day and the TKM extract at 100 mg/kg/day significantly increased the time spent in the target quadrant. It suggests an enhancement in cognitive function. 

The locomotor test (LT) measures the overall movement of animals as differences in activity or drug might be able to interfere with behavioral model testing. The results indicated no statistically significant difference in LT among the experimental groups, as shown in Figure 4 (for the detailed statistical analysis, see Appendix A).

### 2.3. Effects of TKM Extract on Uterus Weight and Volume and Serum 17β-Estradiol (E2) Levels

OVX group exhibited a significant reduction in both uterine weight and volume and a decline in the amount of E2 in the serum compared to the group that underwent sham surgery. After eight weeks, the injection of E2 (1 μg/kg/day) to OVX mice as a form of hormone replacement therapy produced a notable rise in the weight and volume of the uterus, as well as an increase in the concentration of E2 in the serum. In OVX mice that were administered, TKM extract did not produce a statistically significant increase in uterine weight and volume. Additionally, as shown in Figure 5 (for the detailed statistical analysis, see Appendix A), the level of serum E2 remained unchanged.

### 2.4. Effects of TKM Extract on Oxidative Stress and Antioxidant Enzyme Activities in the Brains

The concentration of malondialdehyde resulting from lipid peroxidation, which indicates oxidative damage, was significantly higher in the hippocampus and frontal cortex of OVX mice compared to the sham-operated control mice. Furthermore, OVX mice exhibited notable decreases in SOD and CAT activities. The administration of E2 daily and supplementation with TKM extract at a dose of 100 mg/kg/day effectively prevented the occurrence of lipid peroxidation caused by OVX. Treatments also restored SOD and CAT activities in both brain regions. TKM extract at a dose of 50 mg/kg/day also restored these activities, although in a lower dose-dependent manner, as shown in Figure 6a (for the detailed statistical analysis, see Appendix A), Figure 6b (for the detailed statistical analysis, see Appendix A), and Figure 6c (for the detailed statistical analysis, see Appendix A).

### 2.5. Effect of TKM Extract on Ovariectomized-Induced Changes in the Hippocampus and Frontal Cortex of Gene Encoding ERα, ERβ, Nrf2, Keap1, BDNF, and CREB

Quantitative real-time polymerase chain reaction (RT-qPCR) was used to reveal the expression of genes encoding the estrogen receptor ERα (Figure 7a) (for the detailed statistical analysis, see Appendix A), ERβ (Figure 7b) (for the detailed statistical analysis, see Appendix A), the major regulators of the cytoprotective response to ROS Nrf2 (Figure 7c) (for the detailed statistical analysis, see Appendix A), keap1 (Figure 7d) (for the detailed statistical analysis, see Appendix A), the neurotrophic factors BDNF (Figure 7e) (for the detailed statistical analysis, see Appendix A), and CREB (Figure 7f) (for the detailed statistical analysis, see Appendix A) in both brain regions. There were significant differences in the expression of all tested genes between the OVX receiving vehicle group compared to the sham-operated control mice. Treatment with either TKM extract or E2 restored the expression of these genes in OVX mice. TKM extract up-regulated ERα and Erβ, Nrf2, BDNF, and CREB and down-regulated Keap1 in OVX mice in a dose-dependent manner.

## 3. Discussion

Postmenopausal women experiencing a reduction in estrogen levels are susceptible to neurodegenerative disorders associated with a deterioration in cognitive brain function [8,30]. Previous research has shown that hormone replacement therapy (HRT) not only alleviates a range of symptoms associated with menopause, but also mitigates the risk of developing Alzheimer’s disease post-menopause [13]. Consequently, we used an ovariectomized (OVX) mice model to assess potential anti-dementia-like behaviors and establish the effect of 17B-estradiol. This model is highly regarded for its ability to accurately simulate women with ovarian hormone insufficiency, leading to accelerated aging, brain cell inflammation, and oxidative stress in the brain [10,31,32].

The current findings indicate that the traditional Thai Tri-Kaysorn-Mas (TKM) formula has the potential to prevent senile dementia in menopausal women caused by estrogen deficiency and pathological anomalies in the central nervous system (CNS). The TKM formula consists of three therapeutic herbs: *A. marmelos*, *N. nucifera*, and *J. multifida.* It was found that TKM also has antioxidant activity and inhibits the enzyme acetylcholinesterase [15,16]. Additionally, previous studies revealed that *A. marmelos* contains high concentrations of various phytochemicals, such as aegeline, scopoletin, and imperatorin, which exhibit phytoestrogen activity and neuroprotective effects [17,18,19]. *N. nuciferara* also contains various chemical components, such as flavonoids and alkaloids [20]. Previous studies have demonstrated that constituents like kaempferol, β-sitosterol, quercetin, and nuciferine, have phytoestrogen properties and can potentially improve cognitive functions associated with learning and spatial memory [21,22,23,24,25]. Moreover, *J. multifida* is recognized for its chemical constituents, such as vitexin. Studies have shown that isovitexin produces estrogenic activity and significantly decreases cognitive impairment [26,27,28,29]. Surprisingly, TKM extract shows better suppression of cognitive impairment by decreasing oxidative brain damage [33]. The need for a more thorough examination of the varied biological actions of neuroprotection is motivated by numerous factors [34]. The plant extract was defined as strong in removing free radicals.

It is widely recognized that menopausal women may experience a deficiency in E2, an essential ovarian hormone. This deficiency can potentially result in brain damage and ultimately lead to senile dementia. The current investigation established that the administration of TKM extract improved cognitive impairment-like behavior in OVX mice, as evidenced by their performance in the Y-maze test, novel object recognition test (NORT), and Morris water maze (MWM) test. We found that the removal of ovaries, which causes a decrease in estrogen levels, was a major factor affecting the cognitive function of mice. The administration of E2 and TKM extract (100 mg/kg/day) significantly enhanced non-spatial cognitive performance in the NORT and spatial cognitive performance in the Y-maze test of OVX mice. Both the TKM extract and 17β-estradiol improved the cognitive performance of OVX mice in both the Y-maze test and the NORT test (Figure 1 and Figure 2). The Y-maze test is a behavioral assessment used in rodents to evaluate spatial recognition, working memory, and spontaneous alternation behavior. To verify the results of the two tests, the MWM test (MWMT) was also conducted. This model is commonly employed to evaluate spatial learning and memory [10,31,32]. During the investigation, the OVX mice administered the vehicle was unable to differentiate between learning and recall memory in the test phase, whereas the sham-operated mice showed no alteration in spatial working memory from the training to the test phases. Surprisingly, OVX mice given the TKM extract at a dose of 100 mg/kg/day showed improved spatial working memory, which had been impaired by the ovariectomy. This improvement was similar to the effects of treatment with 17β-estradiol in the MWMT (Figure 3). The current behavioral study provided definitive evidence that daily administration of TKM extract and E2 led to improvements in brain-dependent spatial working memory (assessed using the Y-maze test) and reference memory (assessed using the NORT) in OVX animals, with the enhancements being dose-dependent. The study’s findings suggest that the use of TKM extract may have a preventative impact on cognitive impairment caused by menopause or estrogen deficiency, a significant contributor to the onset of dementia, including Alzheimer’s disease [4]. Behavioral experiments on learning and memory indicate that the TKM extract is likely beneficial in reducing impairments in non-spatial, short-term spatial, and long-term spatial working memory caused by estrogen deficiency in a dose-dependent manner. Moreover, neither of the behavioral trials showed significant enhancement in locomotor activity, suggesting that the TKM extract did not stimulate the central nervous system, which could otherwise lead to a deceptive representation of activity associated with behavioral testing. 

To clarify whether TKM involved the reproductive organ, we found that OVX causes uterus atrophy, while OVX mice receiving E2 had significantly increased uterus weight, volume, and serum E2. TKM extract did not show improvement in uterus weight, volume, and serum E2 (Figure 5). This indicated that TKM extract may improve cognitive impairment and be safe from reproductive side effects. Furthermore, we conducted a study to examine the effects of E2 and the TKM extract on cognitive impairment caused by OVX mice during behavioral trials. The primary emphasis was on understanding the processes behind oxidative stress-induced brain membrane damage. Scientific research indicates that a lack of estrogen raises the likelihood of brain damage, ultimately leading to the death of neurons. The potential mechanism involves the oxidative route, which plays important roles in neurogenesis, neural plasticity, and the ERK1/2 signaling pathway by activating estrogen receptors. This process helps maintain homeostasis in response to oxidative stress and inflammation [9]. Moreover, research indicates that phytoestrogen and antioxidant chemicals have neuroprotective properties that reduce neuronal death caused by oxidative stress in the brain, specifically through the process of lipid peroxidation [35,36]. Based on previous studies, the levels of TBARs (thiobarbituric acid reactive substances), which indicate lipid peroxidation, were noticeably higher in the hippocampus and frontal cortex of OVX mice compared to sham-operated mice. However, when the TKM extract was administered, the increasing TBARs level was effectively prevented in a dose-dependent manner. Remarkably, the TKM extract demonstrated superior inhibition of lipid peroxidation compared to E2 in the brains of OVX mice. Furthermore, our findings align with the notion that OVX-induced cognitive impairment is caused by recurrent stress triggering the excessive formation of free radicals and the inhibition of antioxidant systems, such as superoxide dismutase (SOD) and catalase (CAT) activities. The activities SOD and CAT were dramatically reduced in OVX mice compared to the sham group. The reduced functioning of these enzymes was linked to the buildup of highly reactive free radicals, resulting in harmful consequences such as the deterioration of cell membrane integrity and function. Additionally, numerous reports on the impact of estrogen receptors α and β on the cellular redox system in neuronal membranes support the hypothesis that these cells could be influenced by estrogen. This prompted us to examine if reactive oxygen species (ROS) and reactive nitrogen species (RNS), which contribute to the abnormal functioning of these tissues, might control these receptors [37]. Increasing the cellular metabolic rate stimulates the production of ATP, which in turn leads to the spontaneous formation of superoxide anions (O_2_^•−^) through mitochondrial complexes I and III of the electron transport chain. The rise in the formation of O_2_^•−^ and other radicals, along with the simultaneous suppression of the body’s natural antioxidant defense system, led to the death of neuronal cells. This was particularly evident in the pyramidal cells of the hippocampus, where the endoplasmic reticulum (ER) is located, resulting in additional cognitive deterioration [38]. Unexpectedly, the administration of TKM extract resulted in a notable enhancement of the brain’s oxidative status in the OVX animals. This was evidenced by the increased activities of antioxidant enzymes, closely resembling the effects observed with 17β-estradiol. The results of our study showed that the TKM extract exhibited significant antioxidant activity in both the ORAC and superoxide radical scavenging assays, aligning with previous research.

Several research studies have shown that menopause stimulates the production of free radicals, which leads to oxidative stress. This oxidative stress is associated with the antioxidative potential of the Nrf2/antioxidant responsive element (ARE) signaling pathway and causes a decrease in the levels of enzymatic antioxidants in different brain regions, resulting in altered neurogenesis, impaired hippocampal synapses, and cognitive decline [39,40]. Hence, we explored the potential of the TKM extract and E2 to efficiently modulate the oxidative stress pathway, estrogen receptor, and neuroplasticity processes in the brain. The results indicated that a deficiency in ovarian hormones led to a considerable decrease in the expression levels of ERα and ERβ mRNA in the brain. These findings align with clinical data indicating that the absence of estrogen leads to a decrease in the expression of ER in the brain. According to these findings, the presence of estrogen or TKM from sources outside of the gonads may play a role in the functional significance of the greater presence of ER in the hippocampus when gonadal hormones are not present. ERs can be regulated not just by hormones but also by extracellular signals, such as insulin growth factor-1 (IGF-1), Epidermal growth factor receptor (EGFR), G protein-coupled receptor (GPER), and Tropomyosin receptor kinase B (TrkB), which activate ERs and increase the expression of genes targeted by ERs [41,42,43,44]. Both 17β-estradiol and the TKM extract dramatically increased the expression of all estrogen receptors in both the hippocampus and frontal cortex of the OVX animals. To validate these findings, we examined the estrogen receptor’s impact on the enhancement of CREB and BDNF gene transcription. These genes are crucial in regulating neuronal damage modulation and promoting neuronal survival [45]. Our results showed that OVX caused a decrease in the expression of CREB and BDNF mRNA in both brain areas. However, this effect was reversed by the administration of E2 and the TKM extract. This finding is consistent with the research conducted by Monthakantirat et al. [10], which highlighted the significance of 17β-estradiol in regulating the CREB and BDNF genes. These genes play a crucial role in synaptic plasticity and neuronal survival. Furthermore, multiple studies have revealed that the absence of Nrf2 in mice, a protein that enhances the production of antioxidant proteins and trophic factors such as BDNF, plays a crucial role in cellular reactions to oxidative stress and provides support for neurotrophic functions. These processes may have relevance in the context of cognitive effects induced by ovariectomy [46]. Prior studies have shown that ovarian hormone deprivation in animals that have undergone OVX can potentially elevate the levels of ERα and ERβ through ERK and PI3K/AKT-mediated pathways. Consequently, this may lead to a decrease in the transcription of CREB and BDNF genes while simultaneously increasing the activation of the Nrf2 signaling pathway (ERs/Nrf2 pathways) (Figure 8) [46,47]. Considering this, TKM extract can be seen as a feasible choice for neuroprotection due to the estrogenic action exhibited by certain plants rich in flavonoids, which have the potential to decrease neurodegeneration and neuronal death. Furthermore, glutamate dehydrogenase (GDH) is an enzyme that catalyzes the reversible oxidation of glutamate. It plays a crucial role in brain function, particularly Alzheimer’s and Parkinson’s disease, energy production, and insulin secretion. GDH is regulated by several ligands, such as estrogens, diethylstilbestrol (but not testosterone) [48], and herbal substances such as polyphenols (epicatechin gallate, epigallocatechin gallate, and others) [49]. TKM extract contains various polyphenols and phytochemicals that potentially regulate GDH. Further studies of the glutamate dehydrogenase activity and expression upon administration of TKM are of interest.

## 4. Materials and Methods

### 4.1. Plant Materials and Extractions

The plant materials of the TKM formula, including *A. marmelos* L. fruit (ABH34), *N. nucifera* Gaertn. Stamen (ABH15), and *J. multifida* L. bark (ABH35), were prepared at a ratio of 1:1:1 and dried at Chao Phya Abhaibhubejhr Hospital, Prachinburi Province, Thailand. The finely powdered TKM formula was extracted three times by macerating with 95% ethanol (1:4) at room temperature for 3 days. Filtrates were concentrated using a rotary evaporator (Buchi, Essen, Germany). The extract was kept at −20 °C for the duration of the experiment. The filtrate was freeze-dried (Labcondo, Kansas, MO, USA) after evaporating until it dried as a powder. Moreover, we confirmed the absence of ethanol by checking ethanol using GC (Hewlett-Packard, Wilmington, DE, USA) It has not been commonly used for ethanol extract previously. This research will help develop pharmaceutical preparation in the future.

### 4.2. Determination of Antioxidant Activity by Oxygen Radical Absorbance Capacity (ORAC)

The ORAC assay was carried out according to the method of Dudonné et al. [50]. The ORAC assay measures the fluorescent signal emitted by a probe, which is quenched in the presence of reactive oxygen species (ROS). Antioxidants like Trolox (a reference standard) counteract ROS, thereby restoring the fluorescent signal. This method uses AAPH (2,2-azobis (2-amidinopropane) dihydrochloride) to produce a peroxyl free radical upon thermal decomposition. The TKM extract (25 µL) was tested with 150 µL fluorescent reagent and 25 µL AAPH in 96-well black plates incubated in 37℃. Fluorescence was measured at excitation 485 nm and emission 538 nm.

### 4.3. Determination of Superoxide Anion (O_2_^•−^) Radicals Scavenging Activity

A modified approach from Pirunkaset et al. [51] was used to create superoxide radicals. The reaction solution consisted of 258 µM nitro blue tetrazolium (NBT) in 10 mM phosphate buffer, pH 7.4, 50 µL of 996 µM nicotinamide adenine dinucleotide phosphate (NADH) solution, and 5 µL of the sample solution in DMSO and was performed at various concentrations. The reaction was started by adding 50 µL of 16.2 μM phenazine methosulfate (PMS) solution in 10 mM sodium phosphate buffer pH 7.4 to the mixture and incubating at 25 °C for 5 min. After the incubation, absorbance at 562 nm was measured. As a control, the reaction mixture without PMS was employed (A0). The samples (A1) were introduced to the reaction mixture, which included O_2_^•−^, which inhibited the NBT reduction. A0 − A1 reflected the drop in O_2_^•−^ after measuring the absorbance. The percent of superoxide anion radicals scavenging activity (SRSA) was calculated according to the following equation:SRSA% = (A0 − A1/A0) × 100(1)

### 4.4. Animals

The ICR female mice (five-week-old: 20–30 g) were housed in transparent enclosures with unrestricted availability of nourishment and water at the Northeast Laboratory Animal Centre at Khon Kean University in Khon Kean, Thailand. The enclosures were kept at a stable temperature of 22 ± 2 °C, and the humidity was consistently maintained at 45 ± 2%. The mice were exposed to a 12 h diurnal cycle, with the lights illuminated from 06:00 am to 06:00 pm. The Animal Ethics Committee of Khon Kean University approved and validated the method for the use and care of animals (Approved No. AEKKU44/66). 

### 4.5. Surgical Procedure and Drug Administration

To anesthetize before the operating procedure, each mouse was injected with xylazine (5 mg/kg i.p.; L.B.S. Laboratories Ltd., Part, Thailand) followed with Zoletil^TM^ 100 (Tiletamine and Zolazepam 37.5 mg/kg *i.p.*; VIRBAC, 06510 Carros, France) [10]. The mice underwent bilateral ovariectomy via a dorsolateral incision, which was employed to stimulate an estrogen-deprivation condition. The exposed ovary and associated oviduct were removed, and skin incisions were closed. After a 3-day recovery period, mice were randomly divided into 5 groups (*n* = 15): (1) sham group (no OVX) receiving 0.5% SCMC (sodium carboxymethyl cellulose), (2) OVX mice receiving 0.5% SCMC, (3) OVX mice receiving intraperitoneal E2 (1 μg/kg/day, *i.p.*), and (4) and (5) OVX mice receiving TKM extract at 50 and 100 mg/kg/day *p.o.*, respectively, for 8 weeks, as shown in the experiment framework in Figure 9. Although intraperitoneal injection is considered a parenteral route of administration, the pharmacokinetics of compounds administered intraperitoneally are more alike to those seen after oral administration because the primary route of absorption is into the mesenteric vessels, which drain into the portal vein and pass through the liver [52]. The administration lasted for 1 h on the days specifically designated for behavioral testing preceding the behavioral evaluation. 

### 4.6. Y-Maze Test

The Y-maze test evaluated the spatial memory of animals, which relies on the hippocampus. The apparatus consisted of three black polyethylene arms with similar dimensions: 3.8 cm width at the base, 12.5 cm width at the top, 40.4 cm length, 18.9 cm height, and 60° orientations. During a 5 min testing period, mice were placed on one arm, and the number of arm entries was manually recorded [31,32]. A solution containing 70% ethanol was used to cleanse the Y-maze after the task to prevent the gathering of scent signals between sessions. The percentage of alternation between all three arms without repetition was calculated according to the following equation:%Alternation = [(Number of alternations)/(Total arm entries − 2)] × 100(2)

### 4.7. Novel Object Recognition Test (NORT)

The working memory was evaluated using NORT and was performed as described previously [31,32]. The apparatus included a square enclosure measuring 50 × 50 × 50 cm. Before the test, the mice were given 15 min to habituate the test box in the absence of the object. There was a 30 min interval between the sample and test portions of the NORT. The sample phase experiment included mice seeing two comparable objects positioned at a predetermined distance in the corner of a test box for 5 min. Each mouse engaged in exploration by smelling, touching, or directly encountering objects. The duration of exploration for each object was recorded. A single analogous object was substituted throughout the test. The investigation of this object lasted for 5 min. To prevent the gathering of scent signals between sessions, the test box arena and object were cleaned using a solution of 70% ethanol. The total time spent exploring each familiar object (TF) and novel object (TN) was analyzed. The discrimination index was calculated as a percentage using the following equation:%Discrimination index = [(TN − TF)/(TN + TF)] × 100(3)

### 4.8. Morris Water Maze Test (MWM)

The MWM test assesses animal spatial and working memory [31,32]. The MWM is a black circular pool filled with water and divided into four quadrants with a submerged platform. Mice underwent four trials daily for five days of training. Each trial began with a mouse in one of four quadrants 90° apart along the pool’s circumference, face to the tank wall, and permitted to swim to the buried platform, which remained stable throughout training. The mouse spent 10 s on the platform before being moved to the chamber with high walls that blocked the view for 60 s. Escape latency was recorded for each attempt to reach the hidden platform. After the four experimental trials, mice were thermally insulated for an hour and then returned to their regular housing environment. Before interpreting results, daily data were averaged over four trials. A single 60 s probe experiment was undertaken to test the mice’s ability to locate the platform after it was removed from the pool. Time spent in the target quadrant (Q1) and the other quadrants (Q2–Q4) of the pool was assessed and compared across groups.

### 4.9. Locomotor Test (LT)

The Y-maze test was conducted to highlight whether each of the treatments had any impact on the locomotor activity or behavior of the mice. A mouse was positioned on one arm and given unrestricted freedom to investigate for 5 min. The whole arm entries were recorded manually.

### 4.10. Measurement of Serum 17β-Estradiol Levels

The cardiac puncture was conducted roughly 20 h after the completion of the behavioral test. This procedure was carried out with thiopental sodium (Anesthal^®^, Jagsonpal Pharmaceutical Ltd., Gurugram, India) (80 mg/kg, *i.p.*) for each mouse. A blood sample of approximately 1 mL was collected and then centrifuged at a speed of 3000× *g* at a temperature of 21 °C for 20 min. The supernatants were collected and stored at −80 °C until E2 analysis. The serum levels of E2 were determined by using the Elecsys Estradiol III following the manufacturer’s guidelines (Cobas^®^, Mannheim, Germany) with electrochemiluminescence immunoassay analyzers [10,31,32].

### 4.11. Dissection of Uterus and Brain Tissue

The act of euthanasia was employed as a means of sacrificing animals. The uterus and brain (hippocampus and frontal cortex) were surgically removed and afterward stored at a temperature of −80 °C until they were utilized. The weight and volume of the uterus were measured for each animal group.

### 4.12. Measurement of Lipid Peroxidation Activity in the Brain

Peroxidation occurs when free radicals remove electrons from cell membrane lipids, notably in the brain, causing oxidative damage. Malondialdehyde (MDA) is a common lipid maker for oxidative stress. Thiobarbituric acid reactive substances (TBARs) tests assess malondialdehyde, which is formed by peroxidation of polyunsaturated fatty acids. According to Butterfield et al. [53]. MDA was used to evaluate TBARs in homogenized hippocampus and frontal cortex brain tissues. After homogenizing brain tissue, trichloroacetic acid was added. The mixture was centrifuged at 8000× *g* for 10 min at 4 °C. The supernatant was incubated with 0.8% TBA at 100 °C for 15 min. The reaction produced a pink TBA-MDA complex. Absorbance was measured at 550 nm. Brain MDA was measured using the MDA standard curve. Picomoles of malondialdehyde per milligram of protein were reported [54].

### 4.13. Measurement of Enzymatic Antioxidant Activities

The enzyme known as superoxide dismutase (SOD) facilitates the process of dismutation, wherein the superoxide anion (O_2_^•−^) is converted into hydrogen peroxide and molecular oxygen. The enzyme catalase (CAT) facilitates the enzymatic breakdown of hydrogen peroxide (H_2_O_2_) into water and oxygen. The presence of O_2_^•−^ and H_2_O_2_ results in the oxidation of biological components, resulting in oxidative damage [31,32]. The cellular defense against oxidative damage involves the enzymatic activities of SOD and CAT, which are responsible for the removal of O_2_^•−^ and H_2_O_2_, respectively. This study aimed to assess the activity of the antioxidant enzymes SOD and CAT in the hippocampus and frontal cortex. To obtain 20% brain homogenates, both brains were homogenized using cold phosphate buffer (5 mM, pH 7.4). The activities of SOD and CAT were quantified using the commercially available assay kits, specifically the 19160 SOD Assay Kit and CAT100 Catalase Assay Kit from Sigma-Aldrich, located in St. Louis, MO, USA. The normalization of antioxidant enzyme activities was conducted by dividing the measured values by the total protein concentration in the sample, which was calculated using the Bradford method [55].

### 4.14. Quantitative Real-Time Polymerase Chain Reaction (RT-qPCR)

The mRNA expression of mouse ERα, ERβ, Nrf2, Keap1, BDNF, and CREB in the brains (hippocampus and frontal cortex) was measured using real-time PCR. The extraction of total RNA from both brains was performed using TRIzol^®^ (Thermo Fisher Scientific Inc., San Jose, CA, USA) following the directions provided by the manufacturer. First-stand cDNA synthesis is achieved using oligo (dT) primers and SuperScript III reverse transcriptase from Thermo Fisher Scientific Inc., located in San Jose, CA, USA. The RT-qPCR analysis uses the SsoAdvanced™ Universal SYBR^®^ Green Supermix (Biorad, Hercules, CA, USA). MArcrogen, a company based in Seoul, South Korea, produced the following primers. The amplification process was conducted using gene-specific PCR primer sets, as shown in Table 2. Following amplification, a melting curve study was conducted for each gene. The expression levels of GAPDH mRNA were utilized as a reference standard for normalizing the data. The data were represented using relative expressions.

### 4.15. Statistical Analysis

The data were presented as the mean ± standard error of the mean (SEM). The data were subjected to statistical analysis using one-way analysis of variance (ANOVA), followed by Tukey’s test for conducting multiple comparisons between several groups. Statistical significance was attributed to differences with *p* < 0.05. Data analysis was conducted using IBM^®^ SPSS^®^ Statistics 29 (IBM Corp.^©^, Armonk, NY, USA).

## 5. Conclusions

This finding demonstrates the novel pharmacological effect of Tri-Kaysorn-Mas extract, which is that the long-term administration of TKM extract enhances cognitive function. TKM potentially improved cognitive impairment associated with estrogen deficiency. TKM exhibited multiple mechanisms of action through various pathways, such as enhancing the oxidative stress response, activating estrogen receptors, promoting neurotrophic factors, and boosting antioxidant activity, ultimately enhancing vata (the wind element of our body). Moreover, TKM extract may be beneficial because it expresses estrogenic function at the molecular level in the brain without showing action in the reproductive organ, unlike hormone replacement therapy. Furthermore, some mechanisms such as GDH, the hypothalamic–pituitary–adrenal axis, and quality control data, are crucial for the next study.

## Figures and Tables

**Figure 1 pharmaceuticals-17-01182-f001:**
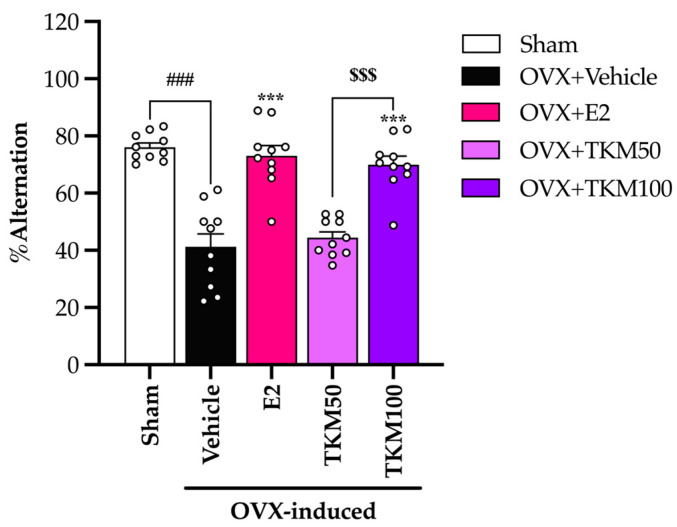
The effect of TKM extract on cognitive performance in sham and OVX mice using the Y-maze test. Statistical significance were analyzed by using one-way ANOVA (Tukey’s test) for multi-comparisons. Each column represents the mean ± SEM (*n* = 10 animals per each group). ^###^ *p* < 0.001 vs. the sham group. *** *p* < 0.001 vs. the OVX group. ^$$$^ *p* < 0.001 between TKM treatment.

**Figure 2 pharmaceuticals-17-01182-f002:**
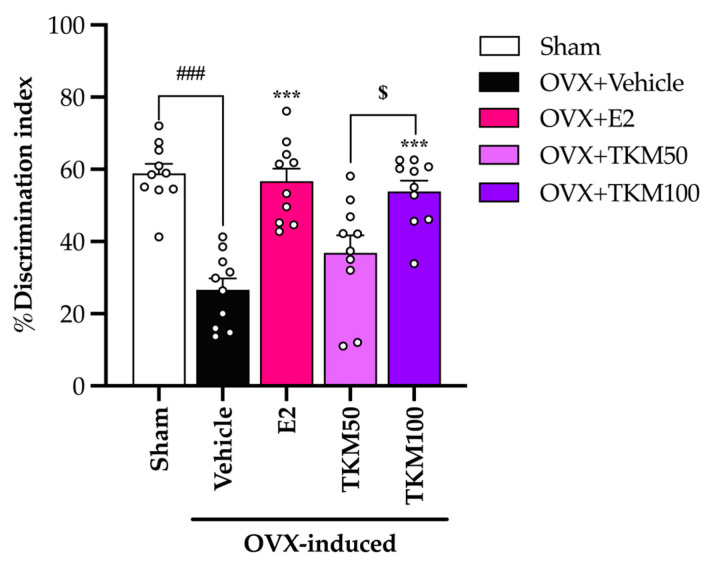
The effect of TKM extract on object recognition memory in the novel object recognition test (NORT). Statistical significance were analyzed by using one-way ANOVA (Tukey’s test) for multi-comparisons. Each experimental group’s percentage of discrimination index values is expressed as the mean ± SEM (*n* = 10 animals per each group). ^###^ *p* < 0.001 vs. the sham group. *** *p* < 0.001 vs. the OVX group. ^$^ *p* < 0.05 between TKM treatment.

**Figure 3 pharmaceuticals-17-01182-f003:**
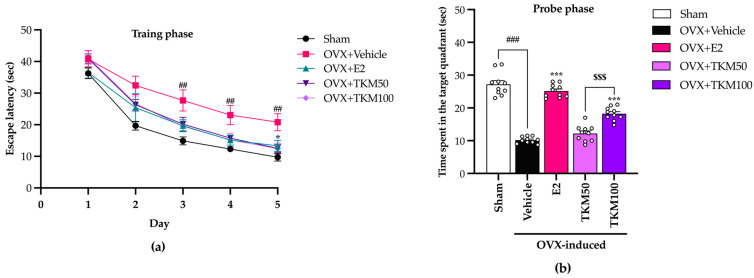
The effect of TKM extract on Morris water maze test (MWM) performance of OVX mice. (**a**) The learning achievement of mice was analyzed during the training phase. Statistical significance were analyzed by using one-way ANOVA (Tukey’s test) for multi-comparisons. The data represent the mean ± SEM of escape latency, four trials/day for five days. (**b**) Memory retrieval performance was examined in the probe phase. The data represent the mean time spent in the target quadrant ± SEM. (*n* = 10 animals per each group). ^##^ *p* < 0.01, ^###^ *p* < 0.001 vs. the sham group. * *p* < 0.05, *** *p* < 0.001 vs. the OVX group. ^$$$^ *p* < 0.001 between the TKM treatment.

**Figure 4 pharmaceuticals-17-01182-f004:**
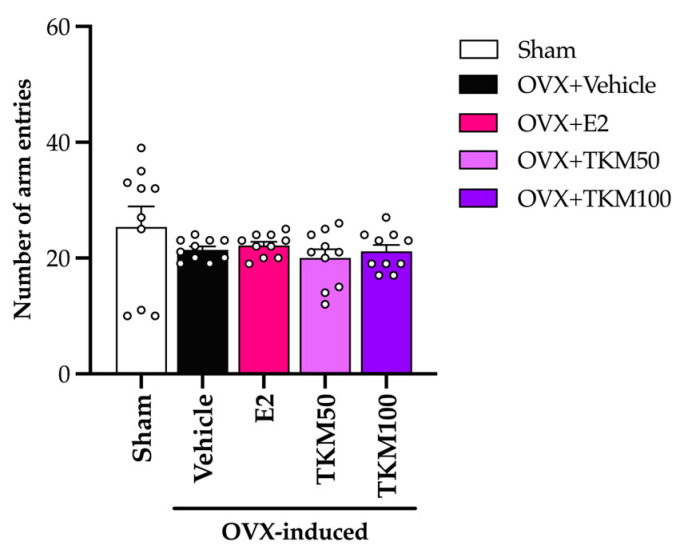
The effect of TKM extract in locomotor test. Statistical significance were analyzed by using one-way ANOVA (Tukey’s test) for multi-comparisons. Each experimental group is expressed as the mean ± SEM (*n* = 10 animals per each group).

**Figure 5 pharmaceuticals-17-01182-f005:**
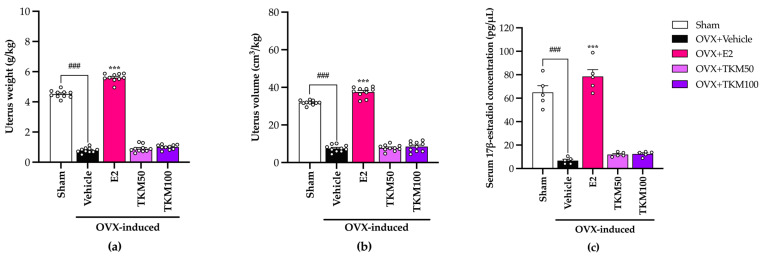
The effect of TKM extract on the uterus and the concentration of serum 17β-estradiol in OVX mice. (**a**,**b**) are the potential and effect of TKM extract on uterus weight and uterus volume, respectively (*n* = 10 animals per each group). (**c**) Serum E2 concentration is a hormone replacement therapy effect (*n* = 5 per each group). Statistical significance were analyzed by using one-way ANOVA (Tukey’s test) for multi-comparisons. The data represent the mean ± SEM. ^###^ *p* < 0.001 vs. the sham group. *** *p* < 0.001 vs. The OVX group.

**Figure 6 pharmaceuticals-17-01182-f006:**
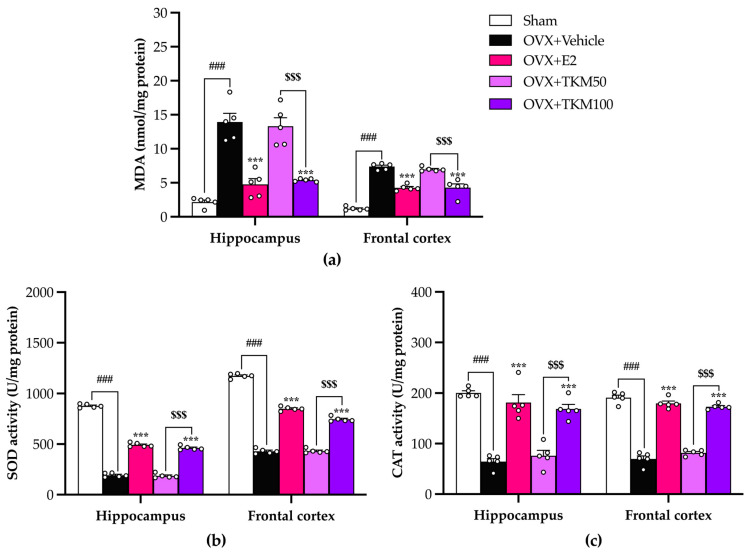
Effect of TKM extract on oxidative damage and antioxidant enzyme activities. (**a**) was the effect of TKM extract on oxidative damage in the hippocampus and frontal cortex. (**b**) was the effect of TKM extract on SOD enzyme activity in the hippocampus and frontal cortex. (**c**) was the effect of TKM extract on CAT enzyme activity in the hippocampus and frontal cortex. Statistical significance were analyzed using one-way ANOVA followed by Tukey’s test for multi-comparisons. The values given were the mean ± SEM (*n* = 5 animals per each group). ^###^ *p* < 0.001 vs. the sham group. *** *p* < 0.001 vs. the OVX group. ^$$$^ *p* < 0.001 vs. the TKM treatment.

**Figure 7 pharmaceuticals-17-01182-f007:**
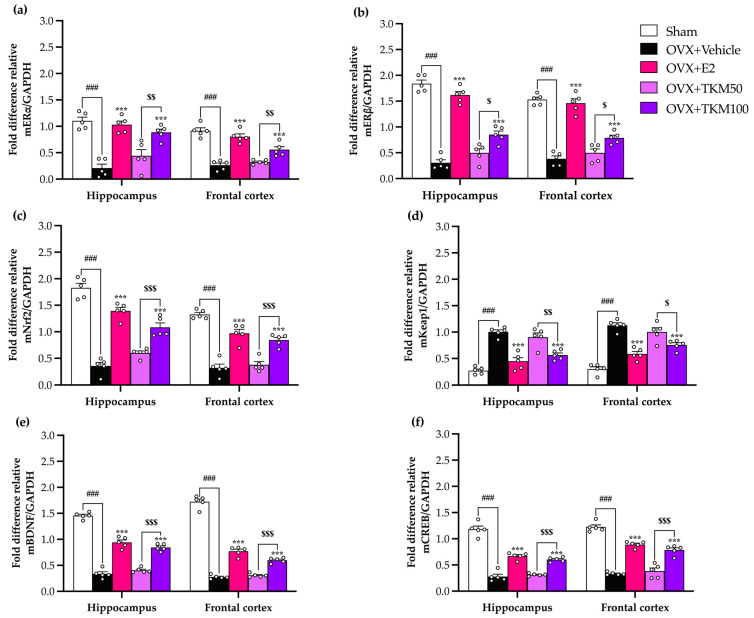
Effects of TKM extract on ovariectomy-induced alternations in ERα (**a**), ERβ (**b**), Nrf2 (**c**), Keap1 (**d**), BDNF (**e**), CREB, and (**f**) mRNA expression in the hippocampus and frontal cortex. The expression level of each mRNA was assessed by RT-qPCR and expressed as a ratio of each mRNA to GAPDH mRNA. Statistical significance were analyzed by using one-way ANOVA (Tukey’s test) for multi-comparisons. Each data column represents the mean ± SEM. (*n* = 5 animals per each group). ^###^ *p* < 0.001 vs. the sham group. *** *p* < 0.001 vs. the OVX group. ^$^ *p* < 0.05, ^$$^ *p* < 0.01, ^$$$^ *p* < 0.001 between the TKM treatment.

**Figure 8 pharmaceuticals-17-01182-f008:**
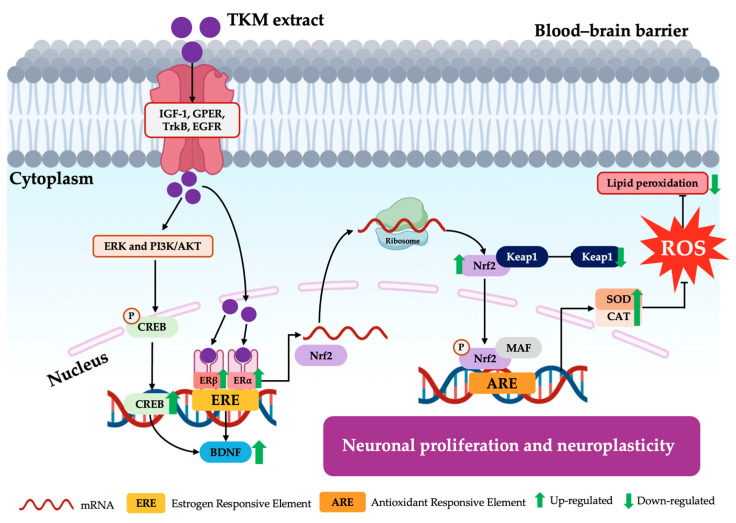
Proposed multi-mechanism of TKM extract on cognitive function in the hippocampus and frontal cortex. TKM extract entry into the central nervous system and absorption occurs via the high-density lipoprotein (HDL) that appears in the Blood–Brain Barrier (BBB). The extract crosses BBB via the multi-receptors in the brain such as IGF-1, GPER, TrkB, and EGFR, to efficiently activate estrogen receptors, neuroplasticity upstream markers (P13K/AKT family), and oxidative stress markers (ERs/Nrf2 pathways). The results show that TKM extract can improve the CREB, BDNF, ERα, and ERβ mRNA expression through this neurogenesis pathway and activate the Nrf2 signaling pathway, which has protection against oxidative stress via Nrf2 and Keap1 mRNA. Furthermore, TKM extract can enhance antioxidant enzymes such as SOD and CAT enzymes. Interestingly, TKM extract can ameliorate lipid peroxidation in the brain. Reactive oxygen species (ROS); superoxide dismutase (SOD); catalase (CAT); phosphoinositide 3-kinases (PI3K); protein kinase B (Akt); extracellular signal-regulated kinases (ERK); cAMP-response element binding protein (CREB); brain-derived growth factor (BDNF); estrogen receptor-α (ERα); estrogen receptor-β (ERβ); nuclear factor erythroid 2-related factor 2 (Nrf2); Kelch-like ECH-associated protein 1 (Keap1).

**Figure 9 pharmaceuticals-17-01182-f009:**
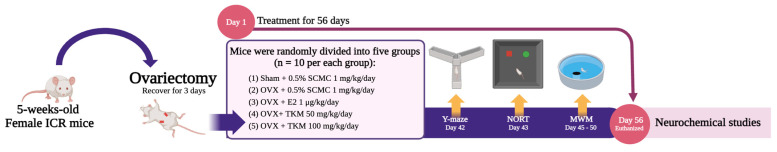
Scheme of the experimental framework. Mice underwent bilateral ovariectomy. Three days after recovery, they were administered intervention for 56 days. Behavioral tests were tested from week six. After behavioral tests, mice were euthanasia, the hippocampus, frontal cortex and serum were collected for neurochemical studies.

**Table 1 pharmaceuticals-17-01182-t001:** Effect of TKM extract on in vitro studies.

Extract	Tri-Kaysorn-Mas (TKM)
ORAC assay (μM Trolox equivalent/g extract)	1.83 ± 0.02
Superoxide radical scavenging (IC_50_) mg/mL	0.59 ± 0.01

**Table 2 pharmaceuticals-17-01182-t002:** Real-Time PCR primer sequences.

Genes		Primer Sequences	Product Length	References
*GAPDH*	*sense*	5′-AACACAGTCCATGCCATCAC-3′	452 bp	[32]
	*antisense*	5′-TCCACCACCCTGTTGCTGTA-3′		
*ER* *α*	*sense*	5′-GACCAGATGGTCAGTGCCTT-3′	166 bp	[56]
	*antisense*	5′-ACTCGAGAAGGTGGACCTGA-3′		
*ERβ*	*sense*	5′-CAGTAACAAGGGCATGGAAC-3′	204 bp	[56]
	*antisense*	5′-GTACATGTCCCACTTCTGAC-3′		
*Nrf2*	*sense*	5′-CAGTGCTCCTATGCGTGAA-3′	109 bp	[57]
	*antisense*	5′-GCGGCTTGAATGTTTGTC-3′		
*Keap1*	*sense*	5′-CATCCACCCTAAGGTCATGGA-3′	291 bp	[58]
	*antisense*	5′-GACAGGTTGAAGAACTCCTCC-3′		
*BDNF*	*sense*	5′-GACAAGGCAACTTGGCCTAC-3′	334 bp	[10]
	*antisense*	5′-CCTGTCACACACGCTCAGCTC-3′		
*CREB*	*sense*	5′-TACCCAGGGAGGAGCAATAC-3′	183 bp	[10]
	*antisense*	5′-GAGGCAGCTTGAACAACAAC-3′		

## Data Availability

The original contributions presented in the study are included in the article/Appendix A, further inquiries can be directed to the corresponding author/s.

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
