# Peer review of "Effectiveness of Tri-Kaysorn-Mas Extract in Ameliorating Cognitive-like Behavior Deficits in Ovariectomized Mice via Activation of Multiple Mechanisms"

_pharmaceuticals, 2024, doi:10.3390/ph17091182_

Round 1

Reviewer 1 Report

Comments and Suggestions for Authors

The manuscript "Effectiveness of Tri-Kaysorn-Mas Formula Extract in Ameliorating Cognitive-Like Behavior Deficits in Ovariectomized Mice via Activation of Multiple Targets" describes an interesting study on a traditional medical herbal preparation. The paper plan has a few features and even sentences in common with a recently published one (https://www.mdpi.com/1420-3049/27/13/4310), cited here too.

I highly recommend a correction of the title. Using the word "formulation" instead of "formula extract" or removal of the word "formula" are proposed.

An abbreviation for E2 as 17β-estradiol is provided only in line 467 (too late). It is better to use only "17β-estradiol" in the text and figures.

The description of Materials and Methods should include a statement, if the procedure used for the TKM maceration and further manipulations is also commonly used for the humans. A reference to a protocol is missing here.

Does the TKM extract contain alcohol? How do the mice behave upon TKM consumption? The TKM is likely bitter, so a more careful description of its administration to mice is needed.

Administration of estradiol and TKM differs, but there are no animal groups to control for the effects of i.p. injections, as well as the controls without OVX but with the TKM or estradiol administration. This is an important limitation of the study, which should be described either in the discussion or as a separate part.

line 49 - "aging females" please specify the species.

Lines 55-56 - the sentence should be made clear to the international audience. Please try explain the words semha, pitta, vata or the phrase in general.

Line 90 - see the comment to the title.

Chapter 2.1 and corresponding method description - It's unclear how a gram of extract corresponds to the initial herbal material. The description should include how many grams of the extract was obtained from the particular mass of herbal material. Similarly, the TKM concentration in uM (Table 1) is not clear.

Line 154. Correct the sentence and related text: "This result suggests that OVX-induced learning and memory impairments."

All figures, including Fig.9 should contain number of animals/samples used (as n=...). When the number of experimental points is n=5-10, each point should be visible on the graphs. Using ANOVA one should confirm the normality of the data. Please add information about the normality tests, exclusions and outliers, if present.

Fig. 7 - the caption should be corrected ("mHouse keeping genes" is weird, especially since you use GAPDH and this is stated in the caption).

Fig. 8 caption should be modified - remove the "(created with Bio Render)" and describe it in methods.

The scheme (Fig. 8) is very speculative, so at least each participant of the scheme should be discussed with references or excluded.

Of interest is the link between the cognitive deficits and the regulation of glutamate dehydrogenase (GDH) by estrogens, but not testosterone (https://doi.org/10.3390/ijms25084341). Moreover the protein is known as a target for herbal drugs such as epicatechin gallate and others (doi: 10.1074/jbc.M111.268599). TKM can highly likely regulate GDH too. Furthermore, GDH is important not only for the brain function but also for the insulin secretion. These data extend the context of the study and are particularly important for the discussion of the current paper. Further studies of the glutamate dehydrogenase activity/expression upon administration of TKM are of interest.

Comments on the Quality of English Language

Comments on the Quality of English include the comments to the title, line 154 and lines 55-56.

Author Response

Response to Reviewer 1’s comments

Response: Thank you very much for kindly spending your precious time reviewing and commenting on our manuscript. Your comments are valuable comments to improve our manuscript. The responses are below.

The manuscript "Effectiveness of Tri-Kaysorn-Mas Formula Extract in Ameliorating Cognitive-Like Behavior Deficits in Ovariectomized Mice via Activation of Multiple Targets" describes an interesting study on a traditional medical herbal preparation. The paper plan has a few features and even sentences in common with a recently published one (https://www.mdpi.com/1420-3049/27/13/4310), cited here too.

Response: Thank you very much; that publication also belongs to our research group.

Comments 1:  I highly recommend a correction of the title. Using the word "formulation" instead of "formula extract" or removal of the word "formula" are proposed.

Response: We have revised following your suggestion.

Comments 2:  An abbreviation for E2 as 17β-estradiol is provided only in line 467 (too late). It is better to use only "17β-estradiol" in the text and figures.

Response: We have deleted 17β-estradiol (Line 442).

Comments 3:  The description of Materials and Methods should include a statement, if the procedure used for the TKM maceration and further manipulations is also commonly used for the humans. A reference to a protocol is missing here.

Response: We have revised following your suggestions.

Comments 4:  Does the TKM extract contain alcohol? How do the mice behave upon TKM consumption? The TKM is likely bitter, so a more careful description of its administration to mice is needed.

      Does the TKM extract contain alcohol?

Response: We used freeze-dried after evaporating until it dried as a powder. 

                  How do the mice behave upon TKM consumption? The TKM is likely bitter, so a more careful      

                   description of its administration to mice is needed.

Response: Animals behave normally. Because TKM was subjected by gavage deeply at the base of the tongue to the pharynx, there were no problems with oral (retching).

Comments 5:  Administration of estradiol and TKM differs, but there are no animal groups to control for the effects of i.p. injections, as well as the controls without OVX but with the TKM or estradiol administration. This is an important limitation of the study, which should be described either in the discussion or as a separate part.

Response: Absorption of material delivered intraperitoneally is typically much slower than for intravenous injection. Although intraperitoneal delivery is considered a parenteral route of administration, the pharmacokinetics of substances administered intraperitoneally are more similar to those seen after oral administration, because the primary route of absorption is into the mesenteric vessels, which drain into the portal vein and pass through the liver.

Lukas G, Brindle SD, Greengard P. 1971. The route of absorption of intraperitoneally administered compounds. J Pharmacol Exp Ther 178:562–566. 

Turner PV, Brabb T, Pekow C, Vasbinder MA. Administration of substances to laboratory animals: routes of administration and factors to consider. J Am Assoc Lab Anim Sci. 2011 Sep;50(5):600-13. 

Becuase, E2 is a pure substance, we selected i.p. as the route of administration, while TKM extract and vehicle were administered by p.o. Moreover, the animals received long-term treatment, and the small difference in absorption between routes of administration may not be as involved as in single-dose treatment.

Comments 6:  line 49 - "aging females" please specify the species.

Response: Female means humans, non-human primates, and rodents. Therefore, we can study the menopause effect from mice to humans.

From reference 5.          Gasbarri, A.; Tavares, M.C.H.; Rodrigues, R.C.; Tomaz, C.; Pompili, A. Estrogen, Cognitive Functions and Emotion: An Overview on Humans, Non-Human Primates and Rodents in Reproductive Years. Reviews in the Neurosciences 2012, 23, doi:10.1515/revneuro-2012-0051.

Comments 7:  Lines 55-56 - the sentence should be made clear to the international audience. Please try explain the words semha, pitta, vata or the phrase in general.

Line 90 - see the comment to the title.

Response: We have revised the following reviewer’s suggestions.

“In Thai traditional medicine, the phrase "the blood will go, and the wind will come" is of great significance in understanding women's health, particularly during menopause. "Blood will go" refers to the reduction of semha and pitta, indicating a decrease in the menstrual period and hormone levels. This leads to the aggravation of vata, or the arrival of wind, which acts as the dysfunction of the blood circulatory system.”

Comments 8:  Chapter 2.1 and corresponding method description - It's unclear how a gram of extract corresponds to the initial herbal material. The description should include how many grams of the extract was obtained from the particular mass of herbal material. Similarly, the TKM concentration in uM (Table 1) is not clear.

Response: We have recalculated and changed the unit to mg/kg and explained how to do the calculation as follows.

Comments 9:  Line 154. Correct the sentence and related text: "This result suggests that OVX-induced learning and memory impairments."

Response: We have cut this sentence since we have to revise the written style to be only the result.

Comments 10:  All figures, including Fig.9 should contain number of animals/samples used (as n=...). When the number of experimental points is n=5-10, each point should be visible on the graphs. Using ANOVA one should confirm the normality of the data. Please add information about the normality tests, exclusions and outliers, if present.

Response: We have revised following the reviewer’s suggestion.

Comments 11:  Fig. 7 - the caption should be corrected ("mHouse keeping genes" is weird, especially since you use GAPDH and this is stated in the caption).

Response: We have revised following the reviewer’s suggestion.

Comments 12:  Fig. 8 caption should be modified - remove the "(created with Bio Render)" and describe it in methods.

The scheme (Fig. 8) is very speculative, so at least each participant of the scheme should be discussed with references or excluded.

Response: We have revised the following reviewer’s suggestions.

Comments 13:  Of interest is the link between the cognitive deficits and the regulation of glutamate dehydrogenase (GDH) by estrogens, but not testosterone (https://doi.org/10.3390/ijms25084341). Moreover the protein is known as a target for herbal drugs such as epicatechin gallate and others (doi: 10.1074/jbc.M111.268599). TKM can highly likely regulate GDH too. Furthermore, GDH is important not only for the brain function but also for the insulin secretion. These data extend the context of the study and are particularly important for the discussion of the current paper. Further studies of the glutamate dehydrogenase activity/expression upon administration of TKM are of interest.

Response: Thank you very much for your kind advice. We are also continuing to study of the anti-anxiety and anti-depression activity of TKM, too.

Reviewer 2 Report

Comments and Suggestions for Authors

On average, women perform better than men on measures of verbal memory, beginning as early as post-puberty. However, women's advantage for verbal memory performance is reduced with menopause. Therefore, the theme of presented manuscript is current and significant. The Authors evaluated the effect of Thailand traditionally used formula – Tri-Kaysorn-Mas (TKM) on cognitive deficits in a mouse model of menopause.

Therefore, in my opinion current manuscript concerns an extremely important and contemporary issue that increasingly affects the functioning of all organisms. Moreover, the manuscript fully meets the requirements for research articles and the Pharmaceuticals journal.

The abstract is a brief summary of all elements of the manuscript. The introduction presents all necessary information and ends with a well-formulated goal of the experiments. The methodology and results are written in a precise, understandable way. Numerous figures and table presenting the experimental results additionally increase the value of the manuscript. Additionally, figures showing the experiment scheme make it easier to understand. The discussion explains all the research conducted by the Authors. Conclusions well suport discussed results. In my opinion, the manuscript is well and reliably written.

My comments are intended only to increase the value of the presented manuscript. I only suggest:

The description of the results contains too much information about the performed tests. They should be deleted, leaving only the results.

In my opinion, statistical comparison of the OVX+TKM50 and OVX+TKM100 groups is not needed. Both should be compared to OVX+vehicle. The dose-dependent effect is obvious.

How do the Authors explain such small time intervals between subsequent behavioral tests. Why weren't they longer?

How many animals were kept in cages? This is very important from the point of view of the stress they experience, which may affect the test results.

The Authors do not provide the values of the ANOVA analysis, this should be supplemented.

In the discussion, the Authors should pay attention to the dose-dependent effect of the extract and whether it may cause side effects similar to those of hormone replacement therapy. Does this extract have any beneficial advantage over hormone therapy?

There is no limitations section. It should be included.

Author Response

Response to Reviewer 2’s comments

On average, women perform better than men on measures of verbal memory, beginning as early as post-puberty. However, women's advantage for verbal memory performance is reduced with menopause. Therefore, the theme of presented manuscript is current and significant. The Authors evaluated the effect of Thailand traditionally used formula – Tri-Kaysorn-Mas (TKM) on cognitive deficits in a mouse model of menopause.

Therefore, in my opinion current manuscript concerns an extremely important and contemporary issue that increasingly affects the functioning of all organisms. Moreover, the manuscript fully meets the requirements for research articles and the Pharmaceuticals journal.

The abstract is a brief summary of all elements of the manuscript. The introduction presents all necessary information and ends with a well-formulated goal of the experiments. The methodology and results are written in a precise, understandable way. Numerous figures and table presenting the experimental results additionally increase the value of the manuscript. Additionally, figures showing the experiment scheme make it easier to understand. The discussion explains all the research conducted by the Authors. Conclusions well suport discussed results. In my opinion, the manuscript is well and reliably written.

Response: Thank you very much for kindly spending your precious time reviewing and commenting on our manuscript. Your comments are valuable comments to improve our manuscript. The responses are below.

My comments are intended only to increase the value of the presented manuscript. I only suggest:

Comments 1:  The description of the results contains too much information about the performed tests. They should be deleted, leaving only the results.

Response: We have revised following your suggestion.

Comments 2:  In my opinion, statistical comparison of the OVX+TKM50 and OVX+TKM100 groups is not needed. Both should be compared to OVX+vehicle. The dose-dependent effect is obvious.

Response: For this point, we would just like to compare the dose-dependent manner between the OVX+TKM50 and OVX+TKM100 groups.

Comments 3:  How do the Authors explain such small time intervals between subsequent behavioral tests. Why weren't they longer?

Response: By the timeline-design of the behavioral test, we start with the test that does not too much force animals to difficult working memory in this order: Y- maze (let animals walk naturally and freely in the Y-maze trail), NORT(more difficult than Y-maze to learn and remember, but animals still walk naturally and freely) and the last test Morris water maze (it is the difficult task, animals were forced to swim), respectively. Moreover, these protocols are commonly used. So we need not to take long time interval between test.

Comments 4:  How many animals were kept in cages? This is very important from the point of view of the stress they experience, which may affect the test results.

Response: Five animals were in a cage size of 19 x 30 x 13 cm.

Comments 5:  The Authors do not provide the values of the ANOVA analysis, this should be supplemented.

Response: Yes, we evaluated using ANOVA. The detailed statistical analysis is in the supplementary Material.

Comments 6:  In the discussion, the Authors should pay attention to the dose-dependent effect of the extract and whether it may cause side effects similar to those of hormone replacement therapy. Does this extract have any beneficial advantage over hormone therapy? There is no limitations section. It should be included.

Response: Thank you very much for your valuable comment. We have added those messages to the discussion.

Reviewer 3 Report

Comments and Suggestions for Authors

This well-designed study investigates scientifically the basis of clinical use of a medicinal herb. The animal model was explored well, and the findings are sound. The hypothesis was proved, and the data was adequately discussed. However, during my revisions, I identified some points that could be better addressed to enhance the quality of the document, which follows:

The title is not adequate. The “multiple targets” should be at least generally identified.

In the abstract, the doses tested and duration of experimental treatment must be included; Keywords should be replaced with terms that were not used in the title or abstract; the authors could explore the Mesh terms available on the Pubmed library;

Despite providing an interesting hypothesis background, the introduction section is too long and could be reduced to enable an easy reading. Furthermore, objectives should be better described;

Please replace the word sacrifice with euthanized;

Authors must evaluate data normality before statistical testing;

The major fragility of this study is the lack of proper phytochemical characterization of the tested extract; furthermore, the exact geographic localization of the plant specimen source should be informed; it is difficult to draw any robust conclusions without such information. I recognize that the study is well-designed, written, and discussed, but it is a relevant aspect that must be addressed. Even considering a widely used medicinal herb, it is essential to identify the phytochemical profile of the extract in use.

The description of the results should approach only the findings without a general definition or explanation. Please, revise it.

The authors must justify the dose-response curve tested using scientific literature instead of clinical experience. Proper references must be cited to support the following statement: “The TKM dose was calculated based on the clinical dose (4 g/day) administered in the hospital. The conversion of this dose into the appropriate dose for mice was achieved using the equation for the human equivalent dose.”

The AchE activity would be relevant evidence in the context of this study.

It would be interesting to see correlations between behavioral and biochemical or molecular findings.

The caption of Figure 8 must include the meaning of abbreviations. Lastly, Figure 9 would benefit from a brief description of the protocol in its caption. The neurochemical studies could also be addressed in the caption.

The conclusion section is too audacious and must be revised to include only the study's findings. Lastly, perspectives based on potential future experiments are welcome in this section.

Author Response

Response to Reviewer 3’s comments

This well-designed study investigates scientifically the basis of clinical use of a medicinal herb. The animal model was explored well, and the findings are sound. The hypothesis was proved, and the data was adequately discussed. However, during my revisions, I identified some points that could be better addressed to enhance the quality of the document, which follows:

Response: Thank you very much for kindly spending your precious time reviewing and commenting on our manuscript. Your comments are valuable comments to improve our manuscript. The responses are below.

Comments 1:  The title is not adequate. The “multiple targets” should be at least generally identified.

Response: The title means that the Tri-Kayson-Mass Formula can act in multiple mechanisms of action. The word “targets “ may easily be confused. And if we point out the mechanisms in general identified, we are afraid that the title may be too long. So, we change the word “targets” to “mechanisms” Effectiveness of Tri-Kaysorn-Mas Formula Extract in Ameliorating Cognitive-Like Behavior Deficits in Ovariectomized Mice via Activation of Multiple Mechanisms

Comments 2:  In the abstract, the doses tested and duration of experimental treatment must be included;

Response: We have added doses tested and duration in line 24.

Keywords should be replaced with terms that were not used in the title or abstract; the authors could explore the Mesh terms available on the Pubmed library;

Response: Thank you very much for your comment. We learn from you. We have replaced them with the term you suggested in line 37.

Comments 3:  Despite providing an interesting hypothesis background, the introduction section is too long and could be reduced to enable an easy reading. Furthermore, objectives should be better described;

Response: We have shortened sentences and cut some sentences. Moreover, objectives have been described more.

Comments 4:  Please replace the word scarified with euthanized;

Response: We have revised line 494.

Comments 5:  Authors must evaluate data normality before statistical testing;

Response: We have evaluated data normality. Could you please see the supplements?

Comments 6:  The major fragility of this study is the lack of proper phytochemical characterization of the tested extract; furthermore, the exact geographic localization of the plant specimen source should be informed; it is difficult to draw any robust conclusions without such information. I recognize that the study is well-designed, written, and discussed, but it is a relevant aspect that must be addressed. Even considering a widely used medicinal herb, it is essential to identify the phytochemical profile of the extract in use.

Response: The phytochemical characterization and validation of the test extract was done using the HPLC method. However, these data are not shown in this manuscript since during the manuscript submission process in another journal. Briefly, result show in the following table

Compounds

Contents (mg/g extract) (n=3)

A. marmelos

Gallic acid

0.03 ± 0.00

Scopoletin

0.17 ± 0.02

Imperatorin

20.25 ± 0.01

N. nucifera

Gallic acid

0.59 ± 0.01

Kaempferol-3-O-glucoside

Vitexin

Nuciferine

10.57 ± 0.02

6.36 ± 0.02

2.84 ± 0.02

J. multifida

Gallic acid

0.72 ± 0.00

Scopoletin

0.003 ± 0.00

Vitexin

0.43 ± 0.01

TKM formula

Gallic acid

1.56 ± 0.54

Kaempferol-3-O-glucoside

5.75 ± 0.03

Scopoletin

0.05 ± 0.00

Vitexin

1.43 ± 0.00

Imperatorin

Nuciferine

8.68 ± 0.07

1.67 ± 0.02

We have indicated voucher specimens of all plant materials for the voucher specimens.

Comments 7:  The description of the results should approach only the findings without a general definition or explanation. Please, revise it.

Response: We have revised following your suggestion.

Comments 8:  The authors must justify the dose-response curve tested using scientific literature instead of clinical experience. Proper references must be cited to support the following statement: “The TKM dose was calculated based on the clinical dose (4 g/day) administered in the hospital. The conversion of this dose into the appropriate dose for mice was achieved using the equation for the human equivalent dose.”

Response: Thank you for your suggestion on this point. However, the design for this research really aims to prove scientific evidence of this Traditional Medicine. So we calculated from this equation.

Moreover, we also reviewed the literature to evaluate the concentration range of active chemical markers in the following table. However, in my opinion, explaining the rationale of the test dose is not necessary.

Comments 9:  The AchE activity would be relevant evidence in the context of this study.

Response: Thank you for your suggestion. However, the AchE activity was tested, and data will be shown in another manuscript with amyloid- b study. I can show you preliminary results.

Comments 10:  It would be interesting to see correlations between behavioral and biochemical or molecular findings.

Response: We will study and discuss in next manuscript.

Comments 11:  The caption of Figure 8 must include the meaning of abbreviations. Lastly, Figure 9 would benefit from a brief description of the protocol in its caption. The neurochemical studies could also be addressed in the caption.

Response: We have added the information following your suggestion.

Comments 12:  The conclusion section is too audacious and must be revised to include only the study's findings. Lastly, perspectives based on potential future experiments are welcome in this section.

Response:  We have revised the conclusion.

Round 2

Reviewer 1 Report

Comments and Suggestions for Authors

While the authors have made a partial revision also answering the questions, not all of them were addressed. That means, when a question is raised by a reviewer, it would also likely be raised by other readers. Thus, you must modify the text, so not only the reviewer could receive your answer, but all the readers would understand the text properly. For example, I couldn't find the answer to my "comment 3". Similarly, the answer to "comment 4" should be incorporated to the text.

The comment 5 points an important limitation of the study, which should be stated (as "limitations of the study"). You can't just ignore it and provide the references only to a reviewer. Add them to your paper instead.

Comment 6 - of course I've seen the reference, but the sentence would look better, if you modify it as I've requested.

The modification of Fig. 8 caption is still not OK. As I've said, the figure is very speculative. Discuss each participant of the scheme (in the "Discussion") or eliminate the elements of the scheme, which are not used in your text.

Discussion of the information raised in "comments 13" as a couple of sentences added to your paragraphs devoted to Fig. 8. would be of interest to the audience too. Especially since you add ROS and metabolic enzymes such as SOD and catalase here. 

Author Response

Response to Reviewer 1’s comment on Round 2

Comments and Suggestions for Authors

While the authors have made a partial revision also answering the questions, not all of them were addressed. That means, when a question is raised by a reviewer, it would also likely be raised by other readers. Thus, you must modify the text, so not only the reviewer could receive your answer, but all the readers would understand the text properly. For example, I couldn't find the answer to my "comment 3". Similarly, the answer to "comment 4" should be incorporated to the text.

Response: Thank you very much for your kind comment and suggestion. We revised comments to comments as follows.

Comments 3:  The description of Materials and Methods should include a statement, if the procedure used for the TKM maceration and further manipulations is also commonly used for the humans. A reference to a protocol is missing here.

Response: Since TKM is commonly used by decoction, infusion, or powder, we designed our experiment using ethanol extraction and developed the quality control by chemical markers (data not shown) in ethanol extract. It has not been commonly used for ethanol extract previously. We expect that this research will be useful for developing pharmaceutical preparation in the future. Therefore, we focused on ethanol extraction (like the title of manuscript). So, we have no reference for the commonly used TKM ethanol extract. However, I have explained the commonly used in lines 95-96.

Comments 4:  Does the TKM extract contain alcohol?

Response: We used freeze-dried after evaporating until it dried as a powder. Moreover, we checked ethanol using GC. There is no ethanol peak, as follows.

                  How do the mice behave upon TKM consumption? The TKM is likely bitter, so a more careful      

                   description of its administration to mice is needed.

Response: I have mentioned how to administer following a guideline for the care and use of animals (IACUC) Line 473-474.

The comment 5 points an important limitation of the study, which should be stated (as "limitations of the study"). You can't just ignore it and provide the references only to a reviewer. Add them to your paper instead.

Comments 5:  Administration of estradiol and TKM differs, but there are no animal groups to control for the effects of i.p. injections, as well as the controls without OVX but with the TKM or estradiol administration. This is an important limitation of the study, which should be described either in the discussion or as a separate part.

Response: We have added the following sentences in lines 469- 473. “Although intraperitoneal injection is considered a parenteral route of administration, the pharmacokinetics of compounds administered intraperitoneally are more alike to those seen after oral administration because the primary route of absorption is into the mesenteric vessels, which drain into the portal vein and pass through the liver”

Lukas G, Brindle SD, Greengard P. 1971. The route of absorption of intraperitoneally administered compounds. J Pharmacol Exp Ther 178:562–566. 

Turner PV, Brabb T, Pekow C, Vasbinder MA. Administration of substances to laboratory animals: routes of administration and factors to consider. J Am Assoc Lab Anim Sci. 2011 Sep;50(5):600-13. 

Comment 6 - line 49 - "aging females" please specify the species.

 Of course I've seen the reference, but the sentence would look better, if you modify it as I've requested.

Response: We have revised the sentences. (line 49-50)

The modification of Fig. 8 caption is still not OK. As I've said, the figure is very speculative. Discuss each participant of the scheme (in the "Discussion") or eliminate the elements of the scheme, which are not used in your text.

Discussion of the information raised in "comments 13" as a couple of sentences added to your paragraphs devoted to Fig. 8. would be of interest to the audience too. Especially since you add ROS and metabolic enzymes such as SOD and catalase here. 

Response: We have revised the comment for Figure 8 (caption and discussion). We also discussed GDH.

Reviewer 3 Report

Comments and Suggestions for Authors

Thank you to the authors for their kind reply and for considering my suggestions. This document demonstrates high scientific quality and is well-written and organized. I want to congratulate the authors on their excellent manuscript. All my concerns were addressed and properly resolved.

Author Response

Response to Reviewer 3’s comment on Round 2

Open Review

Quality of English Language

( ) I am not qualified to assess the quality of English in this paper
( ) English very difficult to understand/incomprehensible
( ) Extensive editing of English language required
( ) Moderate editing of English language required
( ) Minor editing of English language required
(x) English language fine. No issues detected

Yes

Can be improved

Must be improved

Not applicable

Does the introduction provide sufficient background and include all relevant references?

(x)

( )

( )

( )

Is the research design appropriate?

(x)

( )

( )

( )

Are the methods adequately described?

(x)

( )

( )

( )

Are the results clearly presented?

(x)

( )

( )

( )

Are the conclusions supported by the results?

(x)

( )

( )

( )

Comments and Suggestions for Authors

Thank you to the authors for their kind reply and for considering my suggestions. This document demonstrates high scientific quality and is well-written and organized. I want to congratulate the authors on their excellent manuscript. All my concerns were addressed and properly resolved.

Response: We are highly appreciative that you are spending precious time reading, commenting, and making suggestions to improve the quality of our manuscript.

Round 3

Reviewer 1 Report

Comments and Suggestions for Authors

Thank you for the revision, there is a great progress, compared to the first version. Still a few efforts and modifications are required. I have to remind again, that all the reviewer's questions and requests should be taken into account in the final version to prevent the same questions and misunderstanding by the readers.

First of all, you've answered the question about the TKM extract very well, so please include it in the text in a short form in Methods. That is, you should add a couple of sentences to the chapter 4.1 (line 426) which will include your statement about TKM: "It has not been commonly used for ethanol extract previously. We expect that this research will be useful for developing pharmaceutical preparation in the future. Therefore, we focused on ethanol extraction...". Here you should also explain that the Filtrates were concentrated (line 425) until complete evaporation of liquid and further used as a powder. (as according to your answer "We used freeze-dried after evaporating until it dried as a powder. Moreover, we checked ethanol using GC. There is no ethanol peak"). In you current version there is no such a detail, which causes unnecessary questions (e.g. about ethanol).

After the sentence finishing with "...which drain into the portal vein and pass through the liver [60]" (line 471) add a sentence stating that the study design does not evaluate or exclude the potential effect of injections and associated stress since the group (3) was the only one receiving intraperitoneal injections.

My other comments have been resolved. There are still sentences and statements which could be improved for the perfection, so I strongly recommend a careful reading and correcting the text at the final steps.

Author Response

Response to Reviewer 1’s comment on Round 3

Comments and Suggestions for Authors

Thank you for the revision, there is a great progress, compared to the first version. Still a few efforts and modifications are required. I have to remind again, that all the reviewer's questions and requests should be taken into account in the final version to prevent the same questions and misunderstanding by the readers.

Response: Thank you very much for your kind comments and suggestions for improving our manuscript. We revised comments to comments as follows.

First of all, you've answered the question about the TKM extract very well, so please include it in the text in a short form in Methods. That is, you should add a couple of sentences to the chapter 4.1 (line 426) which will include your statement about TKM: "It has not been commonly used for ethanol extract previously. We expect that this research will be useful for developing pharmaceutical preparation in the future. Therefore, we focused on ethanol extraction...". Here you should also explain that the Filtrates were concentrated (line 425) until complete evaporation of liquid and further used as a powder. (as according to your answer "We used freeze-dried after evaporating until it dried as a powder. Moreover, we checked ethanol using GC. There is no ethanol peak"). In you current version there is no such a detail, which causes unnecessary questions (e.g. about ethanol).

Response: We added the explanation sentence in chapter 4.1: “The filtrate was freeze-dried after evaporating until it dried as a powder. Moreover, we confirmed the absence of ethanol by checking ethanol using GC. It has not been commonly used for ethanol extract previously. This research will help develop pharmaceutical preparation in the future.”

After the sentence finishing with "...which drain into the portal vein and pass through the liver [60]" (line 471) add a sentence stating that the study design does not evaluate or exclude the potential effect of injections and associated stress since the group (3) was the only one receiving intraperitoneal injections.

Response: At this point, since there are several papers that validate (as the references), therefore these routes of administration were commonly used. So I think it does not to mention.

My other comments have been resolved. There are still sentences and statements which could be improved for the perfection, so I strongly recommend a careful reading and correcting the text at the final steps.

Response: Thank you very much for your kind comments and suggestions for improving our manuscript.  Thank you very much for your kind comments and suggestions for improving our manuscript. We revised some points.
